# HGF-MiLaG: Hierarchical Graph Fusion for Emotion Recognition in Conversation with Mid-Late Gender-Aware Strategy

**DOI:** 10.3390/s25041182

**Published:** 2025-02-14

**Authors:** Yihan Wang, Rongrong Hao, Ziheng Li, Xinhe Kuang, Jiacheng Dong, Qi Zhang, Fengkui Qian, Changzeng Fu

**Affiliations:** 1Sydney Smart Technology College, Northeastern University, Qinhuangdao Campus, Qinhuangdao 066004, China; 202219206@stu.neuq.edu.cn (Y.W.); 202219259@stu.neuq.edu.cn (R.H.); 202219039@stu.neuq.edu.cn (Z.L.); 202319257@stu.neuq.edu.cn (X.K.); 202219071@stu.neuq.edu.cn (J.D.); 202212078@stu.neu.edu.cn (Q.Z.); 2372410@stu.neu.edu.cn (F.Q.); 2Osaka University, Toyonaka Campus, Osaka 560-0043, Japan; 3Hebei Key Laboratory of Marine Perception Network and Data Processing, Northeastern University, Qinhuangdao Campus, Qinhuangdao 066004, China

**Keywords:** emotion recognition, hierarchical graph fusion, mid-late multilevel gender-aware strategy, multi-task learning

## Abstract

Emotion recognition in conversation (ERC) is an important research direction in the field of human-computer interaction (HCI), which recognizes emotions by analyzing utterance signals to enhance user experience and plays an important role in several domains. However, existing research on ERC mainly focuses on constructing graph networks by directly modeling interactions on multimodal fused features, which cannot adequately capture the complex dialog dependency based on time, speaker, modalities, etc. In addition, existing multi-task learning frameworks for ERC do not systematically investigate how and where gender information is injected into the model to optimize ERC performance. To address the above problems, this paper proposes a Hierarchical Graph Fusion for ERC with Mid-Late Gender-aware Strategy (HGF-MiLaG). HGF-MiLaG uses hierarchical fusion graph to adequately capture intra-modal and inter-modal speaker dependency and temporal dependency. In addition, HGF-MiLaG explores the effect of the location of gender information injections on ERC performance, and ultimately employs a Mid-Late multilevel gender-aware strategy in order to allow the hierarchical graph network to determine the proportion of emotion and gender information in the classifier. Empirical results on two public multimodal datasets(i.e.,IEMOCAP and MELD), demonstrate that HGF-MiLaG outperforms existing methods.

## 1. Introduction

With the rapid development of AI technology, the application of emotion recognition in conversation (ERC) in human-computer interaction (HCI) has gained widespread attention. ERC technology is capable of recognizing the emotion of speakers by analyzing signals (i.e., audio, video, text, etc.) in utterance. By recognizing and responding to emotional state of users, machines can provide a more personalized and empathetic interaction experience, deepening the human emotional connection between humans and machines, thus enhancing user experience [1,2]. Additionally, ERC has played a significant role in the fields of opinion mining in social media [3], doctor-patient interaction in clinical practice [4], depression diagnosis [5], and electronic learning environments [6].

In ERC, understanding and utilizing contextual information is particularly important [7]. Capture of speaker dependency and temporal dependency can provide rich contextual understanding. Speaker dependency refers to emotional interactions between individuals in a conversation [8]. This dependency includes changes in not only the individual’s own emotional state, but also emotional interactions between different speakers. For example, as shown in Figure 1, an example of intra-speaker dependency is that in u7, the son expresses frustration, an emotional state that may have been influenced by his own previous utterances (u3, u5). An example of inter-speaker dependency is that in u10, the son expresses concern about his mother’s angry feelings, which may have been influenced by his mother’s angry utterance in u8 and u9. Temporal dependency involves changes in emotional states over time in a conversation. Past utterance influences the trajectory of future utterance, and future utterance may fill in missing information in the past utterance [9]. For example, as shown in Figure 1, ‘frustrated’ in u5 is influenced by ‘sad’ in u4, and it is also supplemented by the information of the emotion state of ‘sad’ of u6 at the next time step. Therefore, the ability to adequately capture speaker dependency and temporal dependency in a conversation is crucial to improve the performance of ERC. HiGRU [10] combined attention and recurrent neural networks to capture neighboring utterance through three main components, but it failed to take speaker dependency into account. To this end, DialogueGCN [11] captured inter-speaker dependence and intra-speaker dependence through a graph network, but it used only textual information and failed to take full advantage of complementarities between different modalities of information, which led to poor performance in complex emotion recognition tasks. Therefore, researchers started to focus on how to integrate information from different modalities using graph networks, proposing cross-modal multi-head attention mechanisms [12,13] and cross-modal feature complementation [14]. However, existing approaches mainly focus on fusing multimodal features and directly modeling the relationships, which cannot adequately capture speaker dependency and temporal dependency within and across modalities. This may lead to a degradation of the overall performance of the model when there are differences in the quality of information across unimodality.

Additionally, adding auxiliary tasks to ERC can improve the performance of the model significantly. Existing multi-task learning, such as the auxiliary tasks in speaker recognition [15], emotion shift detection (ESD-ERC model) [16], facial expression perception (Facial MMT framework) [17], provided additional supervised signals to the ERC model, which helped to improve the model’s understanding and prediction of emotional states. However, most multitask learning fails to utilize gender information effectively, or does not systematically examine how and where gender information is injected into the model to optimize ERC performance. Studies have shown that males and females differ in emotion expression, and these differences are not only reflected in the audio characteristics of speech, such as pitch, intensity, and sound quality, but may also affect the accuracy of emotion recognition [18]. Ignoring gender information can lead to models that fail to model individual differences effectively, thus affecting the personalization and accuracy of emotion recognition.

To address these limitations, this paper introduces Hierarchical Graph Fusion and Mid-Late Gender-aware Strategy (HGF-MiLaG). HGF-MiLaG captures the emotional dynamics of a conversation by constructing Hierarchical Fusion Graph in order to enhance the model’s understanding of semantics and emotions. Specifically, the hierarchical fusion graph is constructed by first constructing unimodal graphs for each modality to capture intra-modal dependency, and then building multimodal graph that enables inter-modal information integration and capture inter-modal speaker dependency, the combination of which is the basis of the hierarchical graph fusion strategy. For the construction of each graph, we model conversations with the help of directed graphs in order to model speaker dependency and temporal dependency. Each node in the graph represents a utterance and the edges represent dependency between utterances. We input the built graph into a graph convolutional network [9] to propagate contextual information. Meanwhile, to address the problem of ignored gender information, we introduce the Mid-Late multilevel gender-aware strategy, and the gender prediction subtask was designed as an auxiliary task to enhance contextual understanding and individual difference recognition for emotion recognition. Considering that mid-stage fusion with feature interaction and fusion at the middle layer of the model can more utilize complementary information between tasks effectively, while late-stage fusion can fuse data from different tasks flexibly. Therefore, in the process of optimizing the multitasking mechanism, we propose an innovative approach of injecting gender-auxiliary information into both the unimodal graph structure with speaker dependency and temporal dependency (mid-stage), and the multimodal graph structure (late-stage), in order to allow the hierarchical graph network to decide how much gender and emotion information to use.

In particular, it should be noted that due to the involvement of gender labels, we need to face up to the potential bias brought about by the use of gender information for modelling. In the data processing process, if there is gender imbalance in the training data or under-representation of emotion data of certain genders, it may lead to bias in the learning process of the model, which in turn may lead to unfair results in practical applications. For example, if the amount of female emotion data is much larger than that of male, the model may identify female emotions more accurately, while the judgement of male emotions is biased. Therefore, in subsequent studies, we will always be highly cautious to circumvent potential biases and ensure the fairness and social value of the model.

General speaking, our main contributions and innovations of this work are as follows:

1. Hierarchical graph fusion: In order to capture and integrate complex dependency in multimodal data, we propose a hierarchical graph fusion strategy. This strategy first constructs three unimodal graphs to capture speaker dependency and temporal dependency within modalities, and then constructs a multimodal graph to represent inter-modal dependency, which ultimately enhances the model’s ability to recognize emotionally relevant features, and provides a richer and more dynamic perspective for the ERC task.

2. Mid-Late multi-Level gender-aware strategy: In order to improve the accuracy of the model in emotion node classification and prediction tasks, we adopt a Mid-Late multi-level gender-aware strategy approach to inject gender information. Taking into account the advantages of the Mid-Late fusion strategy that feature interaction and fusion at the middle layer of the model can utilize the complementary information between tasks more effectively, and the late-stage fusion that can flexibly fuse the data from different tasks, we finally adopt the simultaneous injection of gender-auxiliary information in both the unimodal graph structure with speaker-dependent and temporal-dependent features (mid-stage) and the multimodal graph structure (late-stage) to improve the model’s performance on a emotion classification task. We further explore the effect of the location of gender information injection on ERC performance in Section 5, and the results corroborate our ideas.

We evaluated the proposed model on the IEMOCAP and MELD dialog datasets and compared it with existing methods. The experimental results show that the model has competitive performance over the chosen competitors on both datasets. The rest of the paper is organized as follows: we present related work in Section 2. The details of our proposed methodology are given in Section 3. The experiments are detailed in Section 4. Results and discussions are presented in Section 5, and Section 6 briefly summarizes our work.

## 2. Related Works

### 2.1. Graph Neural Network

In the field of conversational emotion recognition, graph neural networks (GNNs) provide a powerful framework for modeling speaker dependency in conversations due to their excellent relational modeling capabilities. Ghosal et al. [11] applied graph neural networks to ERC for the first time by proposing the DialogueGCN model, which solved the context propagation problem that existed in recurrent neural network (RNN)-based models. However, the method mainly focused on textual information and does not integrate multimodal data. To solve this problem, Hu et al. [12] proposed a multimodal graph convolutional network (MMGCN) based on DialogueGCN, which improves the accuracy of emotion recognition by exploiting the dependency between modalities. However, MMGCN crudely connected the utterance to all other utterances within the modality, which brought additional noise. For this reason, Li et al. [19] proposed a novel multimodal fusion method (GraphMFT) based on graph neural networks, which reduced the noise by constructing multiple heterogeneous graphs and introducing an improved graph attention network. However, the above GNN-based method establishes utterance dependency under a fixed window and thus tends to over-consider contextual information that is weakly or irrelevantly related to the current utterance. Therefore, Gan et al. [20] proposed a model for recognizing emotions in conversations using graph neural networks, supplemented with novel context filters and feature correction mechanisms, which yielded superior performance in the task of conversational emotion recognition. The graph neural network model of HAM-GNN proposed by Fu et al. [21] efficiently models conversational behavior labels by capturing interactions between speakers and contextual semantics, enabling efficient conversational behavior classification. It can be concluded that graph neural networks (GNNs) have made significant progress in the field of ERC. In this paper, we propose the hierarchical graph fusion strategy to adequately capture intra-modal and inter-modal dependency.

### 2.2. Multi-Task Learning

Multi-task learning is an efficient machine learning strategy [22] that enhances the generalization ability of a model by training multiple related tasks simultaneously. The core advantage of this approach is the ability to share knowledge points and utilize the correlation between tasks to enhance the learning efficiency of each task. Ruder [23] pointed out that the use of a multi-task learning approach can simultaneously optimize multiple emotion-related subtasks during the training process, which not only helps to reduce the overfitting problem of the model, but also improves the recognition effect. In addition, collaboration between the tasks also promotes complementary information between modalities and effectively avoids affecting the performance of the whole model due to insufficient data or noise in one modality [24]. A typical example is the multimodal emotion recognition model based on multi-task learning proposed by Wang et al. [25], which improved modal fusion by setting auxiliary tasks to learn more emotionally inclined visual and audio representations. Xue et al. [26] investigated and proposed a self-supervised dynamic fusion model for multi-task multimodal interactive learning, centered on text modality and supplemented by audio modality and video modality, using distribution similarity loss function and heterogeneity loss function to learn commonality characterization and characteristic characterization of modalities. Based on this, multi-task learning is used to obtain the consistency and difference characterization of the modalities. These studies have fully demonstrated the potential of multitask learning in multimodal emotion recognition. In this paper, we further introduce a multitask learning module for Mid-Late multilevel gender-aware strategy to enhance contextual understanding and individual difference recognition for emotion recognition, thus further improving the performance of the model.

### 2.3. Auxiliary Information Fusion

Individual difference modeling is one of the key directions for optimization of ERC, which belongs to the subtask of auxiliary information fusion [27]. In the process of individual difference modeling, the location factor of fusion is crucial, and different fusion locations can significantly affect the model performance. Depending on the location of the fusion, the research methods are primarily divided into early-stage fusion, mid-stage fusion, and late-stage fusion.

Early-stage fusion strategies mainly inject auxiliary information at the input layer [28]. For example, in the field of multimodal emotion analysis, research in Tian et al. [29] improved model performance by incorporating weakly labeled emotion information based on emoji filtering into word vector representations and introducing external feature extraction algorithms. This approach simplifies the subsequent processing by merging data from different modalities right at the data preprocessing stage. However, the disadvantage of early-stage fusion is that it may lead to under-modeling of high-dimensional feature space and complex relationships between modalities [30]. Late-stage fusion strategies merge information from different tasks in the final stage of the model. For example, Zadeh et al. [31] proposed the Multi-Attention Recurrent Network (MARN). MARN recognizes emotions by merging information from different modalities in the final stage of the model. The disadvantage of late fusion is that it may not adequately capture information about interactions between different modalities because each modality is processed independently, which may lead to information loss [32]. The mid-stage fusion strategy, on the other hand, performs feature interaction and fusion at the middle layer of the model [31]. Wu et al. [33] proposed a multimodal emotion recognition method based on the assistance of affective information, which simultaneously improves the performance of emotion classification and emotion recognition tasks by means of joint learning. Specifically, the method encoded modality internal information through a private network layer and achieved mid-stage fusion by jointly learning the main and auxiliary tasks through a shared network layer. The mid-stage fusion strategy provides a way to balance the disadvantages of early-stage and late-stage fusion due to its ability to perform feature interaction and fusion at the middle layer of the model, which can utilize the complementary information between the tasks more effectively, improving the accuracy and robustness of emotion recognition.

Therefore, in the process of optimizing the multitasking mechanism, we propose an innovative approach of injecting both gender-auxiliary information into the unimodal graph structure with speaker dependency and temporal dependency (mid-stage) and the multimodal graph structure (late-stage), which is done in order to allow the hierarchical graph network to decide on the proportion of gender information and emotion information utilized.

## 3. Method

This section describes the proposed method in this paper in detail. The overall architecture of our method is shown in Figure 2, including context encoding, construction of hierarchical graph based on graph networks, injection of gender information and emotion prediction.

### 3.1. Task Definition

In the ERC scenario, each dialog contains M utterances $\left\{u_{1},u_{2},\ldots ,u_{M}\right\}$, and each utterance incorporates information from three modalities: audio ($u_{i}^{a}$), text ($u_{i}^{t}$), and video ($u_{i}^{v}$). The core task of ERC is to assign each utterance in the dialog a emotion label $\left\{y_{1},y_{2},\ldots ,y_{M}\right\}$. Our model exploits the speaker dependency and temporal dependency within and across modalities, and proposes a hierarchical graph fusion strategy. In addition, in order to improve the accuracy of emotion recognition, a Mid-Late multilevel gender-aware strategy is adopted to incorporate gender information as an auxiliary task in the model training.

### 3.2. Unimodal Feature Extraction

AUDIO: We used the OpenSmile toolkit and its IS10 configuration [34] for audio feature extraction. During feature extraction, the window length was set to 25 ms, the step size to 10 ms, and the frames were windowed using a Hamming window. Finally, the acoustic features of each speech are represented as a 1582-dimensional feature vector.

TEXT: We use RoBERTa model [10] to extract the context independent utterance level feature vector. Specifically, we fine-tune the RoBERTa Large model so that each utterance can be generated by the model as a 1024-dimensional feature vector representation.

VISUAL: In line with previous methods [35], we extract visual features using the DenseNet model [36] to generate a feature vector of dimension 342 for each utterance.

### 3.3. Hierarchical Graph Fusion

#### 3.3.1. Context Encoding

To obtain the contextual information of each modality, we refer to DialogueGCN [11] and use bidirectional gated recurrent unit ($\overset{↔}{\mathrm{GRU}}$).(1)$$h_{i}^{\left(a,t,v\right)}=\overset{↔}{\mathrm{GRU}}\left(u_{i}^{\left(a,t,v\right)},h_{i(\pm 1)}^{\left(a,t,v\right)}\right)$$
where $u_{i}^{\left(a,t,v\right)}$ is the context-independent feature representation of utterance i from the audio, textual, and visual modalities, respectively. $h_{i}^{\left(a,t,v\right)}$ denotes the sequential contextual utterance of the output of each modal encoder.

#### 3.3.2. Hierarchical Fusion Graph Construction

We construct directed graphs for each modality to dynamically capture speaker dependency and temporal dependency between utterances. A conversation with M utterances is represented as three unimodal graphs $G^{\left(a,t,v\right)}=\left(V^{\left(a,t,v\right)},E^{\left(a,t,v\right)},T^{\left(a,t,v\right)}\right)$, where $V^{\left(a,t,v\right)}$ stands for vertices, $E^{\left(a,t,v\right)}$ stands for edges, and $T^{\left(a,t,v\right)}$ stands for edge types.

Vertex: Each utterance in a conversation contains three modalities, so each utterance is represented as a vertex $v_{i}^{\left(a,t,v\right)}\in V^{\left(a,t,v\right)}$ in the directed graph of audio, textual and visual modalities, respectively. Where each vertex is initialized with the feature vector $h_{i}^{\left(a,t,v\right)}$ encoded in the corresponding context.

Edge: In a graph, an edge represents a connection between vertices. In our model, it is assumed that each utterance in a set of conversations has a direct dependency on all other utterances in the conversation, i.e., a conversation is constructed as a fully connected graph.

Edge Types: In dialog analysis, the connectivity relations between edges need to be considered in terms of speaker dependency and temporal dependency. Specifically, speaker dependency includes intra-speaker ($S_{intra}$) dependency and inter-speaker ($S_{inter}$) dependency. In this paper, only two speakers (s1 and s2) are considered for conversational emotion recognition, so there are four relationship types for speaker dependency, i.e., speaker 1 self-dependency, speaker 2 self-dependency, speaker 1’s dependency on speaker 2, and speaker 2’s dependency on speaker 1. Temporal dependency considers the order in which utterance appears in a conversation, i.e., there are two types of relations: dependency of present utterance on future utterance and dependency of future utterance on present utterance. We use $\alpha_{i,j}^{\left(a,t,v\right)}\in \left(0,1\right)$ to encode speaker dependency in three unimodal graphs: when there is some speaker dependency between two vertices, $\alpha_{i,j}^{\left(a,t,v\right)}$ is 1; otherwise, $\alpha_{i,j}^{\left(a,t,v\right)}$ is 0. Similarly, we encode the temporal dependence by $\beta_{i,j}^{\left(a,t,v\right)}\in \left(0,1\right)$. Thus for each modality there are at most 8 different relation types $T^{\left(a,t,v\right)}=\left(\alpha_{i,j}^{\left(a,t,v\right)},\beta_{i,j}^{\left(a,t,v\right)}\right)$.

Graph Transformer: We introduce the Graph Transformer model [37] to update the feature vectors of vertices and the weights of edges in unimodal graphs to obtain speaker-dependent and temporal-dependent vertex features. The model incorporates the classical multi-head attention mechanism, making it applicable to graph-structured data.(2)$$h_{i}^{(a,t,v)\prime }=W_{1}^{\left(a,t,v\right)}h_{i}^{\left(a,t,v\right)}+\frac{1}{N_{T}}\sum_{\tau \in T}\sum_{j\in N_{i}}\theta_{i,j}^{\tau (a,t,v)}W_{2}^{\left(a,t,v\right)}h_{j}^{\left(a,t,v\right)}$$
where $W_{n}^{\left(a,t,v\right)}$ is the trainable matrix, $N_{T}$ denotes the total number of relation types of edges, $N_{i}$ denotes the total number of utterances, and $\theta_{i,j}^{\tau (a,t,v)}$ is the attention coefficients computed by the multi-head dot product attention mechanism.(3)$$\theta_{i,j}^{\tau (a,t,v)}=\mathrm{Softmax}\left(\frac{{\left(W_{3}^{\left(a,t,v\right)}h_{i}^{\left(a,t,v\right)}\right)}^{⊤}\left(W_{4}^{\left(a,t,v\right)}h_{j}^{\left(a,t,v\right)}\right)}{\sqrt{D}}\right)$$
where D is the dimension of the model.

Inspired by the studies of Chudasama et al. [37] and Deng et al. [38], we introduce a multi-modal cross-modal attention mechanism to fuse three unimodal features. The multimodal feature is finally represented by $H_{m}$.

Similar to the construction of per-modal graphs, we utilize the output multimodal feature representation $H_{m}$ to construct the multimodal graph structure $G^{m}=\left(V^{m},E^{m},T^{m}\right)$ via speaker dependency and temporal dependency, where $V^{m}$ are the nodes, $E^{m}$ are the edges, and $T^{m})$ are the type of relationship of the edges. Our goal is to utilize the graph structure to more fully integrate the multimodal features of different information dynamics.

Based on the multimodal graph construction, we utilize the graph convolutional network RGCN to deal with different types of relationships between different types of interactions such as intra-speaker and inter-speaker interactions. The capability of RGCN lies in its ability to learn unique representations of different types of relationships, which provides richer contextual information for our model.

First, the multimodal feature representation $H_{m}$ is input to the RGCN, which combines information from neighboring nodes as well as specific relationship types to update the feature representation of a node.(4)$$h_{i}^{m\prime }=\sigma \left(\vartheta_{\mathrm{root}}\cdot h_{i}^{m}+\sum_{\tau \in T}\sum_{j\in N(i)}\frac{1}{\left|N(i)\right|}\vartheta_{\tau }\cdot h_{j}^{m}\right)$$
where $\vartheta_{\mathrm{root}}$ is a learnable parameter of the RGCN for combining the features of the node itself and its neighboring nodes. N(i) is the set of neighboring nodes of node *i* under relation type τ. $\vartheta_{\tau }$ is the learnable weight matrix associated with relation type τ. σ(.) is the ReLU activation function.

After RGCN processing, we use the Weisfeiler-Lehman algorithm [39] to further refine and summarize graph-related features. The WL algorithm encodes the topology of the graph by iteratively updating the feature representations of the nodes, thus generating a feature vector for each node containing information about the global graph structure. The WL processing can increase the model’s ability to perceive the overall structure of the graph.(5)$$m_{i}=W_{\mathrm{Mult}}^{\left(1\right)}\cdot h_{i}^{m\prime }+W_{\mathrm{Mult}}^{\left(2\right)}\sum_{j\in N(i)}\omega_{i,j}\cdot h_{j}^{m\prime }$$
where $\omega_{i,j}$ is the edge weight from source node *j* to target node *i*.

### 3.4. Mid-Late Gender-Aware Emotion Feature Classifier

In this section, we introduce a gender prediction task that aims to integrate gender information as an auxiliary feature into our model. The goal of the auxiliary task is to predict the gender attributes of the dialog participants, thus enriching the model’s understanding of user characteristics. In this way, our model is able to more accurately capture and reflect gender-related emotional features in conversations, which ultimately improves the accuracy of the model’s main task of emotion recognition. To achieve this goal, we design a multi-task learning framework [23] inspired by existing hard parameter sharing strategies [40] and customized for our research goals. In particular, we adopt a Mid-Late multilevel gender-aware approach to inject gender information, which is a way to improve the performance of the model’s emotion classification task by simultaneously injecting gender-auxiliary information in both the speaker-dependent and temporal-dependent unimodal graph structure (mid-stage), and in the fused multimodal graph structure (late-stage). Based on this, we constructed a lightweight multi-task GNN model, HGF-MiLaG, which shares all the convolutional layers between gender prediction and emotion prediction tasks to improve the accuracy of emotion recognition.

#### 3.4.1. Middle Gender-Aware Emotion Classifier

In a unimodal gender-aware emotion classification task, feature vectors $h_{i}^{{\left(a,t,v\right)}^{\prime }}$ of each modal utterance are fed into respective mid-stage gender-aware emotion classifiers to predict emotion labels and gender labels. Specifically, the classifiers first compute the unnormalized emotion or gender category labels $C_{i,emo}^{\left(a,t,v\right)}$ and $C_{i,gen}^{\left(a,t,v\right)}$ via a ReLU activation function, and then apply a Softmax function to obtain the probability distributions $P_{i,emo}^{\left(a,t,v\right)}$ and $P_{i,gen}^{\left(a,t,v\right)}$. Ultimately, the model selects the category with the highest probability as the emotion or gender labels ${\hat{y}}_{i,emo}^{\left(a,t,v\right)}$ and ${\hat{y}}_{i,gen}^{\left(a,t,v\right)}$ for each modal prediction. The features of the two tasks are represented as: (6)$$C_{i,emo}^{\left(a,t,v\right)}=\mathrm{ReLU}\left(W_{e}h_{i}^{{\left(a,t,v\right)}^{\prime }}+b_{e}\right)$$(7)$$P_{i,emo}^{\left(a,t,v\right)}=\mathrm{Softmax}\left(W_{pe}C_{i,emo}^{\left(a,t,v\right)}+b_{pe}\right)$$(8)$${\hat{y}}_{i,emo}^{\left(a,t,v\right)}=\mathrm{argmax}\left(P_{i,emo}^{\left(a,t,v\right)}\right)$$(9)$$C_{i,gen}^{\left(a,t,v\right)}=\mathrm{ReLU}\left(W_{g}h_{i}^{{\left(a,t,v\right)}^{\prime }}+b_{g}\right)$$(10)$$P_{i,gen}^{\left(a,t,v\right)}=\mathrm{Softmax}\left(W_{pg}C_{i,gen}^{\left(a,t,v\right)}+b_{pg}\right)$$(11)$${\hat{y}}_{i,gen}^{\left(a,t,v\right)}=\mathrm{argmax}\left(P_{i,gen}^{\left(a,t,v\right)}\right)$$
where $C_{i,emo}^{\left(a,t,v\right)}$ and $C_{i,gen}^{\left(a,t,v\right)}$ denote the score vectors for emotion or gender categorization after processing by ReLU activation function, $P_{i,emo}^{\left(a,t,v\right)}$ and $P_{i,gen}^{\left(a,t,v\right)}$ are the obtained emotion and gender probability distributions, ${\hat{y}}_{i,emo}^{\left(a,t,v\right)}$ and ${\hat{y}}_{i,gen}^{\left(a,t,v\right)}$ are the emotion and gender labels predicted by the model, which is the most probable category in the probability distribution.

#### 3.4.2. Late Gender-Aware Emotion Classifier

To avoid omitting the initial multimodal representation information from the final emotion output, the processed multimodal representation $m_{i}$ and the unprocessed multimodal representation $h_{i}^{m}$ were concatenated together as a joint multimodal representation input into a late gender-aware emotion classifier, going through the following steps:(12)$$m_{i}^{\prime }=\mathrm{Concat}\left(m_{i},h_{i}^{m}\right)$$(13)$$M^{\prime }=\left[m_{0}^{\prime },m_{1}^{\prime },\ldots ,m_{t}^{\prime }\right]$$

In order to capture the temporal dynamics of emotions in the dialog, i.e., to take into account the influence of the emotional dynamics of previous speeches on the emotions of current speeches in the model, we constructed the module of temporal attention, and input the feature sequences $M^{\prime }$ into the formula of temporal attention to get the attention weights $\xi^{m}$. After processing, the model can reflect the evolution and dynamics of the emotional state in the dialog.(14)$$\xi^{m}=\mathrm{softmax}\left(m_{i}^{\prime T}W_{M}\right)M^{\prime }\cdot M^{\prime T}$$
where $W_{M}$ is a trainable weight matrix used to map features to a new space to capture temporal relationships.

$\xi^{m}$ is fed into the multimodal classifier to obtain the multimodal part of the output $P_{i,emo}^{m}$ and $P_{i,gen}^{m}$.(15)$$C_{i,emo}^{m}=\mathrm{ReLU}\left(\xi^{m}m_{i}^{\prime }+b_{e}\right)$$(16)$$P_{i,emo}^{m}=\mathrm{Softmax}\left(W_{pe}C_{i,emo}^{m}+b_{pe}\right)$$(17)$$C_{i,gen}^{m}=\mathrm{ReLU}\left(\xi^{m}m_{i}^{\prime }+b_{g}\right)$$(18)$$P_{i,gen}^{m}=\mathrm{Softmax}\left(W_{pg}C_{i,gen}^{m}+b_{pg}\right)$$

Ultimately, the probability distribution $P_{i,emo}^{\left(a,t,v\right)}$ for emotion category and $P_{i,gen}^{\left(a,t,v\right)}$ for gender category obtained by the mid-state gender-aware emotion classifier as well as the probability distribution $P_{i,emo}^{m}$ for each emotion category and $P_{i,gen}^{m}$ for each gender category obtained by the late-state gender-aware emotion classifier are combined as the final prediction of each utterance’s emotion label or gender labeling tool.(19)$$P_{i,emo}=P_{i,emo}^{m}+\sum_{k\in \{a,t,v\}}\delta_{k}P_{i,emo}^{k}$$(20)$${\hat{y}}_{i,emo}=\arg\max\left(P_{i,emo}\right)$$(21)$$P_{i,gen}=P_{i,gen}^{m}+\sum_{k\in \{a,t,v\}}\delta_{k}P_{i,gen}^{k}$$(22)$${\hat{y}}_{i,gen}=\arg\max\left(P_{i,gen}\right)$$
where ${\hat{y}}_{i,emo}$ and ${\hat{y}}_{i,gen}$ are the emotion and gender labels predicted by the utterance $u_{i}$ respectively. $\delta_{a,t,v}$ is a pre-determined hyperparameter.

In the Mid-Late multilevel gender-aware strategy, the model learns both emotion and gender features simultaneously by jointly optimizing the loss function, and the parameter update of the model is guided by the jointly optimized loss function, which enables the model to capture the complex interactions between the gender features and the emotion expression, and thus largely improves the overall performance of HGF-MiLaG.

#### 3.4.3. Loss Function

Throughout the process, we use a single loss function to train all classifiers simultaneously. Given that categorical cross-entropy is well-suited for classification tasks and L2 regularization helps mitigate overfitting [41], we employ categorical cross-entropy as the loss function and incorporate L2 regularization to train our model in an end-to-end manner via backpropagation. Since there is a gender task involved, the total loss function is the sum of the loss functions of the gender classification task and the emotion classification task. It is composed as: (23)$$\mathrm{Loss}=\alpha \cdot {\mathrm{Loss}}_{\mathrm{emo}}+\beta \cdot {\mathrm{Loss}}_{\mathrm{gen}}$$
where α is the weight of emotion information and β is the weight of gender information, the sum of the weights is 1. The weights of the two pieces of information are determined through a hierarchical graph network.(24)$${\mathrm{Loss}}_{\mathrm{emo}}^{\left(a,t,v,m\right)}=-\frac{1}{N}\sum_{i=1}^{N}\log P_{i,\mathrm{emo}}^{\left(a,t,v,m\right)}\cdot \left[y_{i,\mathrm{emo}}^{\left(a,t,v,m\right)}\right]+\lambda {∥\Theta ∥}_{2}$$(25)$${\mathrm{Loss}}_{\mathrm{emo}}=\sum_{k\in \{a,t,v,m\}}\theta_{k}\cdot {\mathrm{Loss}}_{\mathrm{emo}}^{k}$$(26)$${\mathrm{Loss}}_{\mathrm{gen}}^{\left(a,t,v,m\right)}=-\frac{1}{N}\sum_{i=1}^{N}\log P_{i,\mathrm{gen}}^{\left(a,t,v,m\right)}\cdot \left[y_{i,\mathrm{gen}}^{\left(a,t,v,m\right)}\right]+\lambda {∥\Theta ∥}_{2}$$(27)$${\mathrm{Loss}}_{\mathrm{gen}}=\sum_{k\in \{a,t,v,m\}}\delta_{k}\cdot {\mathrm{Loss}}_{\mathrm{gen}}^{k}$$
where N is the number of utterances in the conversation, $y_{i,\mathrm{emo}}^{\left(a,t,v,m\right)}$ and $y_{i,\mathrm{gen}}^{\left(a,t,v,m\right)}$ are the base truth labels for a single emotion prediction or gender prediction task respectively. Specifically, $y_{i}=1$ if the emotion or gender type of a sample utterance *i* belongs to the *i*th class, and $y_{i}=0$ otherwise. λ is the L2-regularity weight, Θ is the set of trainable parameters, and θ and δ are the weights of the respective modality (including multimodality) associated with emotion and gender, respectively.

## 4. Experiment

### 4.1. Dataset

We evaluated our proposed HGF-MiLaG in two public datasets: IEMOCAP [42] and MELD [43].

IEMOCAP: This dataset records the performances of 10 actors in 5 binary sessions covering 12 h of audio-visual data as well as textual transcriptions. Each session involved 2 actors and was segmented into individual utterances. Each utterance was labeled with one of the following six emotion labels: happy, sad, neutral, excited, frustrated, and angry.

MELD: This dataset is a large multimodal multi-party affective dialogue dataset that contains more than 13,000 utterance fragments extracted from the classic American TV show “Friends”, which are organized into approximately 1400 dialogues. Each utterance in the dialog is labeled with one of the following seven emotions: neutral, anger, disgust, fear, joy, sadness, surprise.

### 4.2. Baseline

To validate the effectiveness of our proposed hierarchical graph fusion with Mid-Late gender-aware strategy, we compared it with several previous baselines. The baselines include DiagueGCN [11], DiagueCRN [44], MMGCN [12], COGMEN [8], GraphCFC [14], GA2MIF [13], GraphMFT [19], GCCL [45], and HiMul-LGG [46]. The details of these models are shown below.

DiagueGCN improves the accuracy of emotion recognition by constructing a directed graph to capture the dependency between individual speeches in a conversation, including inter-dependency and intra-dependency between speakers, and utilizing graph convolutional networks to propagate the contextual information in these dependency.

DiagueCRN mimics unique human cognitive thinking by designing a multi-round reasoning module in the cognitive stage to iteratively perform intuitive retrieval processes and conscious reasoning processes, thus providing a deeper understanding of the conversational context and identifying the key cues that trigger the current emotion.

MMGCN is a graph convolutional network model that fuses audio, visual, and textual modalities to enable information interaction by constructing graphs and establishing inter-modal edge connections, and injecting speaker embeddings to capture speaker dependency. The model employs a spectral domain graph convolutional network and extends to deep layers to enhance emotion recognition performance.

COGMEN is a cognitive graph-based emotion recognition model focused on modeling the process of human emotion understanding. The model captures the emotional dynamics in a conversation by combining local information (internal and external dependency between speakers) and global information (conversation context) using Graph Convolutional Networks (GCN) and Graph Transformers.

GraphCFC effectively mitigates the heterogeneity gap problem in multimodal fusion by using multiple subspace extractors and the pairwise cross-modal complementation (PairCC) strategy. By extracting multiple edges from the graph, the GNN can capture key contextual and interaction information in the message delivery more accurately. In addition, the model designs the GAT-MLP structure, which provides a new unified framework for multimodal learning.

GA2MIF focuses on contextual modeling and cross-modal modeling by utilizing multi-head directed graph attention networks (MDGATs) and multi-head paired cross-modal attention networks (MPCATs), respectively, and is able to capture the long-term contextual information within modalities and the complementary information between modalities efficiently.

GraphMFT integrates data objects from different modalities by constructing a graph that utilizes multiple improved graph attention networks to capture intra-modal contextual information and inter-modal complementary information.

GCCL is a multimodal emotion recognition framework that captures speaker, temporal, and inter-modal dependency and integrates multimodal information through a graph-based module. The framework includes a emotion consensus learning unit and a consensus-aware unit with an attention mechanism that ensures individual diversity and inter-modal semantic consistency as well as maintains category-level semantic associations across samples.

COLD Fusion quantifies modality-specific probability or data uncertainty to predict emotion through calibration and ordinal latent distribution fusion. It learns unimodal temporal contextual latent distributions by limiting variance and designs softmax distribution matching loss for uncertainty-weighted fusion. The method significantly improves the generalisation performance and robustness of emotion recognition on multiple datasets.

HiMul-LGG employs a hierarchical decision fusion strategy to ensure cross-modal feature consistency and a local-global graph neural network architecture to enhance inter-modality and intra-modality speaker dependency. In addition, HiMul-LGG utilizes a cross-modal multi-head attention mechanism to facilitate inter-modal interactions.

### 4.3. Experiment Setup

In this study, all experiments were run on NVIDIA GeForce RTX 4060 laptop Gpus (NVIDIA Corporation, Santa Clara, CA, USA). The training framework uses PyTorch 2.5.1 (developed by Facebook AI Research Team), Python version 3.9.0, CUDA Toolkit version 12.4 (NVIDIA Corporation, Santa Clara, CA, USA). During training, the Adam optimizer is used to train our network with the dropout rate of 0.1 and the learning rate set to 0.0003. And the batch size is set to 32 for all datasets. For context encoding, the number of units of the GRUs used is set to 160. The hidden dimension of the hidden layer of the hierarchical fusion graph and the hidden dimension of the temporal attention are also set to 160. The weights of the emotion information and the gender information are set to 0.7 and 0.3, respectively. The pre-determined hyper-parameters $\delta_{a,t,v}$ are 0.1, 0.7, and 0.2, respectively. Our evaluation method is the same as that of the chosen baseline article method, where the weighted average F1 scores and the average accuracies are used to evaluate the performance of HGF-MiLaG.

## 5. Results and Discussion

In this section, experimental results will be reported to evaluate the proposed HGF-MiLaG. Firstly, an overall comparison of HGF-MiLaG with all baseline methods will be made. Then, the effects of different components in the ablation experiments on HGF-MiLaG are discussed. Further, we specifically discuss the effect of injecting gender information at different locations on the effectiveness of model implementation. Next, we explore the effects of different time window settings on HGF-MiLaG. Finally, we also performed an error analysis of our method.

### 5.1. Comparison with Baseline Models

Our model obtained optimal weighted average F1 scores on both the IEMOCAP and MELD datasets. Table 1 compares the experimental results of HGF-MiLaG with other baseline models (mentioned in Section 4.2). As can be seen from Table 1, for the IEMOCAP dataset, the accuracy and F1 score of HGF-MiLaG are 70.98% and 71.92%, respectively, which are 0.86% and 1.11% more accurate than the current state-of-the-art models HiMul-LGG and GCCL, respectively, and the F1 scores are 0.80% and 1.73% more accurate, respectively. In addition, we show the corresponding F1 scores for each emotion label in detail in the table, and HGF-MiLaG shows significant improvement on the emotion label of Happy. Notably, the F1 scores of HGF-MiLaG are significantly higher than the current state-of-the-art models GCCL and AVL COLD Fusion for all six emotion categories.For the MELD dataset, the accuracy and F1 scores of HGF-MiLaG are 66.22% and 65.26%, respectively, and similarly, compared to the accuracies of HiMul-LGG and GCCL models by 0.01% and 3.40%, respectively, and F1 scores by 0.08% and 4.98% over the HiMul-LGG and GCCL models, respectively.

Based on these results, the HGF-MiLaG model exhibits superior performance on both datasets, which is mainly attributed to the synergistic effect of hierarchical graph fusion and Mid-Late multilevel gender-aware strategy. The hierarchical graph fusion strategy effectively captures speaker dependency and temporal dependency within and across modalities by constructing unimodal and cross-modal graph structures. This enables the model to better understand the complex modal relationships and temporal dynamics in emotional expressions. The Mid-Late multilevel gender-aware strategy further enhances the model’s contextual understanding of emotion recognition and its ability to recognise individual differences by simultaneously injecting gender-auxiliary information in both the middle and late stages of the model. Gender plays a key role in emotion expression, and there are often differences in the way males and females express their emotions. By introducing gender information, the model is able to better distinguish these differences, thus improving the overall recognition accuracy. The combination of these two approaches enables the model to not only capture rich modal and temporal information, but also deeply understand gender differences in emotion expressions, thus achieving better performance in complex emotion recognition tasks.

### 5.2. Ablation Study

To better understand the role of our model components, we performed ablation experiments on key components of HGF-MiLaG. The results are shown in Table 2.

Effect of MiLaG. We first explored the impact of the Mid-Late multilevel gender-aware strategy on the model’s emotion categorization task. For this purpose, we removed the gender-auxiliary information injection module, and what can be seen is that the accuracy decreased by 1.11% and the F1-weighted average score by 1.01% in the IEMOCAP dataset, and the accuracy decreased by 0.31% and the F1-weighted average score by 0.35% in the MELD dataset. Introducing gender-auxiliary information can provide the model with additional a priori knowledge to help it better capture individual differences in emotion expression. This strategy can enhance the model’s contextual understanding of emotion recognition and make it more accurate in handling emotion classification tasks. This fully demonstrates the effectiveness of introducing gender-auxiliary information in ERC. It can enhance the model’s contextual understanding of emotion recognition and individual difference recognition.

Impact of the MG. We then explored the impact of multimodal graphs on the model emotion classification task. For this purpose, we removed the multimodal fusion graph and relied solely on the three unimodal graph classifiers for emotion classification, in which case we can see a decrease of 5.55% in accuracy and 5.57% in F1-weighted average score for the IEMOCAP dataset, and a decrease of 0.58% in accuracy and 0.37% in F1-weighted average score for the MELD dataset. The reason for this is that there is complementarity and correlation between different modes, for example, intonation and rhythm of speech can enhance the intensity of emotion in text, while visual information (such as facial expressions) can further assist the recognition of emotion. As an effective intermodal information integration tool, multimodal graph can fuse information of different modes and capture intermodal speaker dependency and temporal dependency. This cross-modal fusion can make up for the deficiency of unimodal information and improve the accuracy and robustness of emotion classification. The fact that the classification performance decreases drastically proves the effectiveness of the multimodal map and the introduction of the multimodal map, which enables cross-modal information integration and thus captures inter-modal speaker dependency and temporal dependency.

Impact of UG. another core of HGF-MiLaG is the unimodal graph. It is responsible for modeling intra-modal speaker dependency and temporal dependency. For this reason, we removed the three unimodal graphs, and the final emotion classification results were determined by the multimodal classifier only, as can be seen by the fact that in the IEMOCAP dataset, the accuracy was reduced by 1.97% and the F1-weighted average score by 1.93%, and in the MELD dataset, the accuracy was reduced by 0.19% and the F1-weighted average score was reduced by 0.51%. According to the dynamic theory of emotion, the expression and transmission of emotion are sequential and dependent. Unimodal graphs can capture this dynamic change of emotion by modeling the temporal dependencies within the modes. In addition, modeling speaker dependence can further enhance the accuracy of emotion recognition, since the emotional expression styles and habits of different speakers may differ. The effectiveness of the unimodal graph lies in its ability to dig deep into the complex relationships within the modes and provide richer semantic and structural information for emotion classification. This demonstrates the validity of the unimodal graphs.

### 5.3. Comparison with Different Position of Gender Injection

In this section, we explore the effect of the location of gender injection on the IEMOCAP dataset, i.e., early, middle, late, and mid-late injection, on the model performance. The results in Table 3 clearly show that injecting gender information in the Mid-Late stage significantly improves the accuracy and weighted F1 scores of the model. Specifically, the F1 scores of injecting gender information in the Mid-Late stage are improved by 1.33%, 1.38%, and 1.81% compared to the F1 scores of injecting gender information in the early, middle, and late stages, respectively. This is due to the unique advantages of Mid-Late multilevel gender aware strategies: Middle stage fusion excels in capturing information about mid-level associations between different modalities, effectively balancing the advantages and disadvantages of early and late fusion. While early fusion facilitates the integration of information at the initial stage, it may lead to information loss or excessive smoothing due to premature merging of features from different modalities. On the other hand, late fusion allows for more flexible processing of data from different tasks in later stages of the model, but may not effectively capture inter-modal associations. The Mid-Late multilevel gender-aware strategy exploits mid-level correlation information by combining the advantages of middle stage fusion and late stage fusion, while maintaining the flexibility and robustness of late-stage fusion. This multilevel gender-aware strategy ensures that the model can better adapt to the complexity of multimodal data, thereby improving its overall performance.

To better understand the effect of gender injection location on the model, we used t-SNE to visualize the potential representations learned by different gender information injection locations on the IEMOCAP test set. Figure 3a–c show the visualization results for gender information injected alone in the early, middle, and late stages, respectively, and Figure 3d shows the visualization results for gender information injected simultaneously in the middle and late stages. Compared with injecting gender information at early, middle, and late stages, our Mid-Late multilevel gender-aware strategy has significantly optimized the clustering of the same emotional category and increased the distance and clarity of boundaries between different emotional categories. Specifically, the representations of sad and happy are clearly distinguishable. Moreover, the overlapping areas of excited, frustrated, and neutral with other emotional categories have been significantly reduced. These results show that MiLaG is able to learn structured representations with clustered levels, improving the model’s ability to recognize emotion categories.

### 5.4. Effect of Window Size

In this section, we discuss the effect of different window sizes on the performance of HGF-MiLaG. Figure 4a,b show the changes in accuracy and weighted F1 values of HGF-MiLaG corresponding to different window sizes on the IEMOCAP and MELD datasets, respectively.

IEMOCAP dataset sees a trend that both F1-w scores and accuracy rates show an increasing and then decreasing trend. The highest values of both F1-w score and accuracy are reached at a window size of 10, both of which are 0.712. while the lowest values of these two metrics are found at window sizes of 0 and 1, which are about 0.679 and 0.677, respectively. the model performance is best at a window size of 10. The model’s performance first improves as the size of the contextual window increases, but beyond a certain size, the effect of the model’s enhancement gradually diminishes, the even to the point of showing a downward trend. The possible reason for this is that shorter context windows cannot capture these long-term dependency, while longer windows help the model to capture these dependency, but beyond a certain length, the increased context information may contain too much noise or irrelevant data, which in turn reduces the model’s recognition ability. Specifically, the optimal window on the IEMOCAP dataset is 10.

The same phenomenon also occurs in the MELD dataset, but with a lower degree of variability and more volatility, where the optimal window is 12. The F1-w scores and accuracy of HGF-MiLaG fluctuate over a small range with window size, and the overall scores are low.The F1-w scores do not vary much between a window size of 0 and 20, with the highest value being about 0.651 and the lowest about 0.645.The accuracy similarly does not vary much between a window size of 0 and 20, with the highest value being about 0.664 and the lowest about 0.657.This indicates that in the second graph, the window size has less effect on the model performance and the performance is more stable. The possible reason for this variability is that the MELD dataset is constructed based on dialogues from the Old Friends TV series, where some of the neighboring utterances are not contiguous in the actual scenarios, making the results somewhat uncertain and requiring longer distances for contextual modeling. In this section, we conclude that the number of windows (i.e., the number of nodes) in graph construction has a significant impact on the model, and that choosing appropriate forward and backward time steps can maximize the model performance and improve the model’s ability for emotion classification tasks.

### 5.5. Insight from Output

To further evaluate the performance of the model, we show in Figure 5 the confusion matrix of HGF-MiLaG on the IEMOCAP and MELD test sets, which is used to assess the quality of the model’s prediction output. With the confusion matrix on the IEMOCAP test set (Figure 5a), we observe that the HGF-MiLaG model has some limitations in distinguishing emotional nuances. For example, the model exhibits low discrimination in recognizing anger vs. frustrated and happy vs. excited. This suggests that the model has room for improvement in capturing subtle differences in emotion.

We have further analysed the MELD test set to compare the model output distribution with the true distribution of the dataset. The results show that the model output distribution is highly consistent with the true distribution of the MELD dataset, and there is no data imbalance that causes the model to over-concentrate on outputting dominant labels.

Specifically, although the emotion labels Anger, Disgust and Fear have a small number of samples and can be easily misclassified as the Neutral label with the largest number of samples, the overall prediction results show that the model’s predicted probability distributions of the emotion labels match with the actual distribution of the dataset. For example, the model’s higher prediction probability for the Neutral label matches the actual situation in the MELD dataset where neutral emotion expressions are more common. This is not a flaw in the model, but rather a reflection of its ability to accurately reflect real-world emotion distributions. In real-world emotion analysis scenarios, the distribution of emotions is inherently uneven, and neutral emotion expressions are usually more common, with strong emotions (e.g., anger, disgust, etc.) being relatively less common, which is related to the natural distribution of people’s daily language expressions and emotional states.

For visual presentation, we draw a histogram of the distribution of the model’s output labels (see Figure 6) and compare it with the true distribution of the MELD dataset. As can be seen from the figure, the model predictions are highly consistent with the real distribution, further confirming the model’s adaptability to real-world emotion distribution.

## 6. Conclusions

In this study, we present the innovative HGF-MiLaG model for analyzing conversational emotional states, and the model adequately combines the hierarchical graph fusion and the Mid-Late multilevel gender-aware strategy to improve the performance of the model’s affective classification task. We provide insights into the performance of our model by comparing it to other baselines and related works in the IEMOCAP and MELD datasets suitable for our experiments. The results reported on both datasets establish the superiority of the HGF-MiLaG model performance. The results of the ablation experiments validate the contribution of the key components of HGF-MiLaG to the model, and show that the hierarchical fusion graph is able to capture speaker-dependency and temporal dependency within and across modalities, and that the introduction of gender-auxiliary information in the ERC strengthens the model’s contextual understanding of emotion recognition and the identification of individual differences. Further, we explored the effect of the location of gender information injection on the performance of ERC, and finally adopted the Mid-Late multilevel gender-aware strategy to realize the ability to let the hierarchical graph network have the ability to decide the proportion of emotion and gender information in the classifier by itself. We believe that this study fully reveals the importance of graph-based modeling techniques and the incorporation of gender-auxiliary tasks at optimal locations for emotion recognition tasks, and that our work has taken an important step forward in the field of modeling the dynamics of complex information for multimodal emotion recognition in conversations. In future work, we will continue to push multimodal learning with a focus on solving the previously mentioned problem of similar emotion discrimination, as well as improving HGF-MiLaG to enhance the model’s ability to deal with sample label imbalance. In addition, we would like to apply our model to more practical settings, similar to the multimodal dialog generation domain, etc., to understand the utility of our model. 

## Figures and Tables

**Figure 1 sensors-25-01182-f001:**
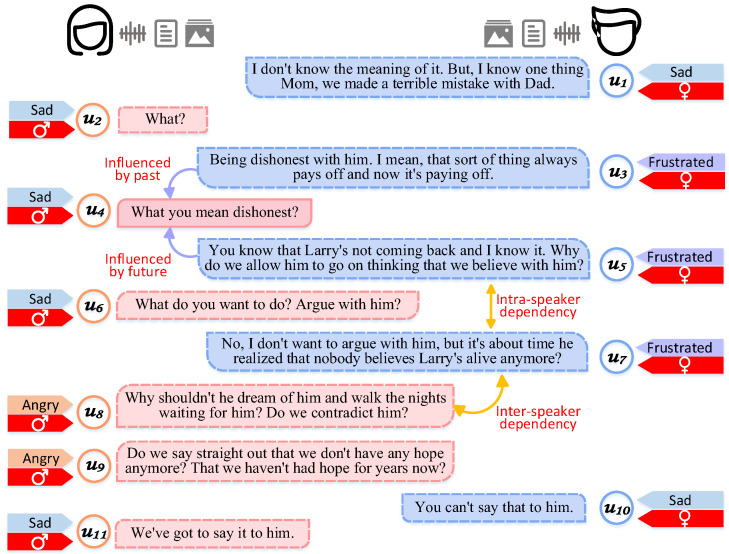
Example of a multimodal dialog scene: The dialog is drawn from textual, audio, and visual modalities. The dialog demonstrates the complexity of communication and emotional relationships faced by the mother and the son when dealing with the loss of a loved one as an intra-familial issue, reflecting speaker-dependent and temporal-dependent relationships.

**Figure 2 sensors-25-01182-f002:**
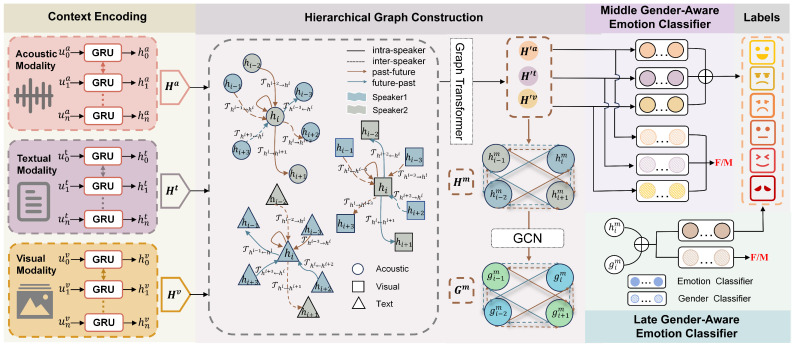
The overall architecture of the proposed HGF-MiLaG, including context encoding, creation of hierarchical graph based on graph networks, auxiliary tasks gender information injection and emotion prediction.

**Figure 3 sensors-25-01182-f003:**
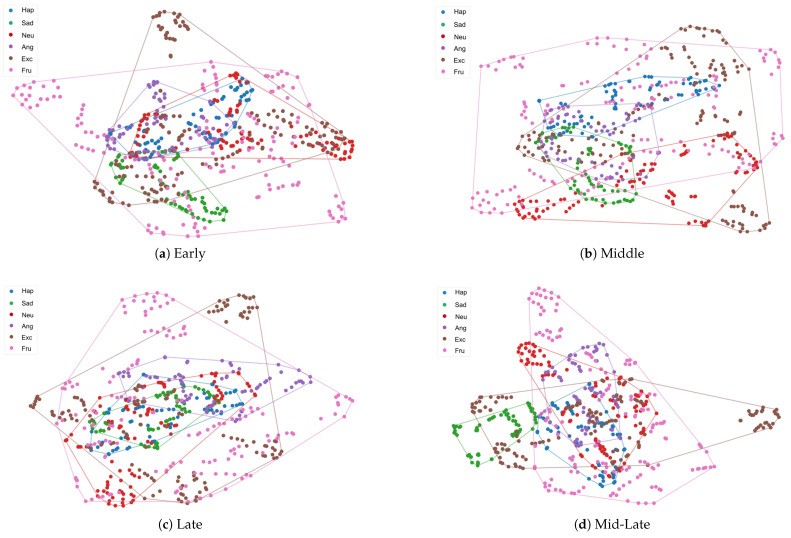
T-SNE visualization of the learned features using different methods on the IEMOCAP test set. In these figures, we use blue, green, red, purple, brown and pink to represent happy, sad, neutral, anger, excited and frustrated, respectively.

**Figure 4 sensors-25-01182-f004:**
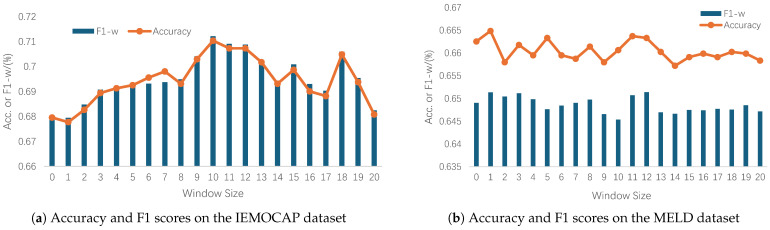
T-SNE visualization of the learned features using different methods on the IEMOCAP test set. In these figures, we use blue, green, red, purple, brown and pink to represent happy, sad, neutral, anger, excited and frustrated, respectively.

**Figure 5 sensors-25-01182-f005:**
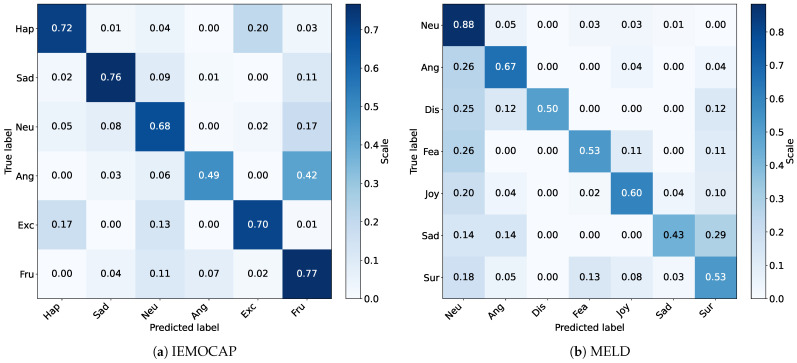
Normalized confusion matrix of HGF-MiLaG on IEMOCAP and MELD test sets. Rows indicate predicted labels and columns indicate true labels.

**Figure 6 sensors-25-01182-f006:**
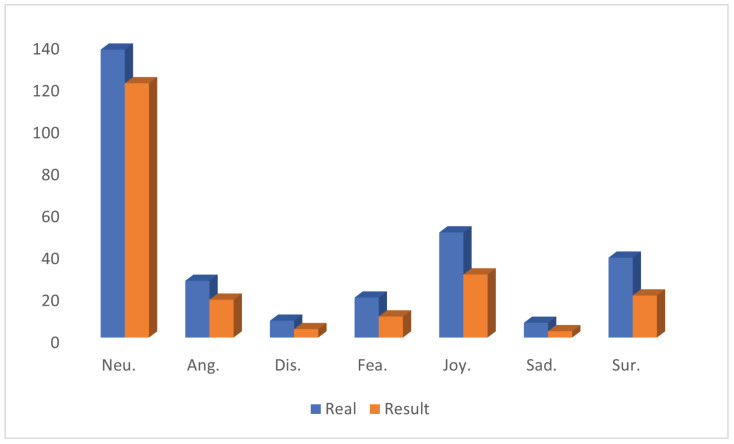
Comparison of model predictions and true emotion label distributions on the MELD test set.

**Table 1 sensors-25-01182-t001:** Comparison with the baseline model: assessment metrics include Acc., F1 and Wa-F1, representing accuracy (%), F1 score (%) and weighted average F1 score (%), respectively. The best performance is indicated in bold.

Model	IEMOCAP								MELD	
	Happy	Sad	Neutral	Angry	Excited	Frustrated	Acc.	Wa-F1	Acc.	Wa-F1
	F1	F1	F1	F1	F1	F1				
DiagueGCN	41.28	82.52	64.33	65.18	73.18	64.46	66.85	66.78	59.58	58.17
DiagueCRN	53.23	83.37	62.96	66.09	75.40	66.07	67.16	67.21	61.11	58.67
MMGCN	37.61	79.84	62.26	** 74.29**	75.00	63.68	67.28	66.67	61.07	57.33
COGMEN	51.90	81.70	68.60	66.00	75.30	58.20	68.20	67.60	-	-
GraphCFC	43.08	**84.99**	64.70	71.35	**78.86**	63.70	69.13	68.91	61.42	58.86
GA2MIF	46.15	84.50	68.38	70.29	75.99	66.49	69.75	70.00	61.65	58.94
GraphMFT	45.99	83.12	63.08	70.30	76.92	63.84	67.90	68.07	61.30	58.37
GCCL	54.05	81.10	70.28	68.21	72.17	64.00	69.87	69.29	62.82	60.28
AVL COLD Fusion	43.70	60.20	48.90	58.40	61.60	57.90	**82.70**	55.10	-	-
HiMul-LGG	53.95	79.92	** 71.66**	67.56	72.00	**68.46**	70.12	70.22	66.21	65.18
HGF-MiLaG	** 59.16**	83.94	70.60	68.60	74.20	66.21	**70.98**	** 71.02**	** 66.22**	**65.26**

**Table 2 sensors-25-01182-t002:** Ablation experiments of HGF-MiLaG: assessment metrics include Acc. and Wa-F1. The best performance is indicated in bold.

Method	IEMOCAP		MELD	
	Acc.	Wa-F1	Acc.	Wa-F1
HGF-MiLaG	**70.98**	**71.02**	** 66.22**	** 65.26**
w/o MiLaG	69.87	70.01	65.91	64.91
w/o MG	65.43	65.45	65.64	64.89
w/o UG	69.01	69.09	66.03	64.75

**Table 3 sensors-25-01182-t003:** Comparison of effects across different gender injection locations on the IEMOCAP dataset. The best performance is indicated in bold.

Position	IEMOCAP	
	Acc.	Wa-F1
Early	69.50	69.69
Middle	69.38	69.64
Late	69.25	69.21
Mid-Late	**70.98**	**71.02**

## Data Availability

The data presented in this study are openly available.

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
