# Peer review of "HGF-MiLaG: Hierarchical Graph Fusion for Emotion Recognition in Conversation with Mid-Late Gender-Aware Strategy"

_sensors, 2025, doi:10.3390/s25041182_

Round 1

Reviewer 1 Report

Comments and Suggestions for Authors

The paper presented the HGF-MiLaG model to improve emotion recognition in conversations (ERC). This step is critical for enhancing human-computer interaction by analyzing multimodal inputs (text, audio, and visual). The paper identifies limitations in prior works, such as insufficient modeling of speaker and temporal dependencies and poor utilization of gender information in ERC.

While the overall paper is well-written, there are still some areas that need to be improved.

-While the model uses gender-aware strategies, it does not discuss ethical implications or potential biases introduced by gender-based modeling.

-There is a risk of model overfitting specially with smaller dataset. Authors may show some results to ensure that result is not overfitted.

-The model struggles with class imbalances in the MELD dataset, which could affect real-world applicability

-Comparison with state of the art would add value to the obtained results

Reviewer 2 Report

Comments and Suggestions for Authors

This paper studies a hierarchical graph fusion for emotion recognition in conversation with mid-late gender-aware strategy. This paper has certain research and application value, but there are some deficiencies need to be improved.

1. Please supplement the reason why the categorical cross-entropy and L2-regularization are used as the loss function in the paper.

2. Please provide additional details on the practical applications of the proposed method and existing methods, as well as the experiment comparison results and analysis.

3. The analysis of the experiment results is rather weak, as well as the description of the experiments.

For the above reasons, we think this paper will be published after revisions in the Sensors.

Round 2

Reviewer 2 Report

Comments and Suggestions for Authors

The authors have revised the shortcomings of the paper, we think this paper can be published in the Sensors.